# Citizen Science-Informed Community Master Planning: Land Use and Built Environment Changes to Increase Flood Resilience and Decrease Contaminant Exposure

**DOI:** 10.3390/ijerph17020486

**Published:** 2020-01-12

**Authors:** Galen Newman, Tianqi Shi, Zhen Yao, Dongying Li, Garett Sansom, Katie Kirsch, Gaston Casillas, Jennifer Horney

**Affiliations:** 1Department of Landscape Architecture and Urban Planning, Texas A & M University, College Station, TX 77843, USA; tianqi0501-@tamu.edu (T.S.); makiyao@tamu.edu (Z.Y.); dli@arch.tamu.edu (D.L.); 2School of Public Health, Texas A & M University, College Station, TX 77843, USA; sansom@sph.tamhsc.edu (G.S.); krkirsch@tamu.edu (K.K.); baker1@tamu.edu (G.C.); 3Epidemiology Program, University of Delaware, Newark, DE 19716, USA; horney@udel.edu

**Keywords:** public health, green infrastructure, landscape performance, resilience, contamination

## Abstract

Communities adjacent to concentrated areas of industrial land use (CAILU) are exposed to elevated levels of pollutants during flood disasters. Many CAILU are also characterized by insufficient infrastructure, poor environmental quality, and socially vulnerable populations. Manchester, TX is a marginalized CAILU neighborhood proximate to several petrochemical industrial sites that is prone to frequent flooding. Pollutants from stormwater runoff discharge from industrial land uses into residential areas have created increased toxicant exposures. Working with local organizations, centers/institutes, stakeholders, and residents, public health researchers sampled air, water, indoor dust, and outdoor soil while researchers from landscape architecture and urban planning applied these findings to develop a community-scaled master plan. The plan utilizes land use and built environment changes to increase flood resiliency and decrease exposure to contaminants. Using a combination of models to assess the performance, costs, and benefits of green infrastructure and pollutant load impacts, the master plan is projected to capture 147,456 cubic feet of runoff, and create $331,400 of annual green benefits by reducing air pollution and energy use, providing pollution treatment, increase carbon dioxide sequestration, and improve groundwater replenishment. Simultaneously, there is a 41% decrease across all analyzed pollutants, reducing exposure to and transferal of toxic materials.

## 1. Introduction

Many neighborhoods located along the Houston Ship Channel, including Manchester, TX, have been documented as having excess risks of exposure to acute pollution, emergency chemical spills and incidents, and high-impact natural and manmade disaster events [1,2,3,4]. These exposures have been linked to poor health outcomes including cancer clusters in both children (brain, leukemia, glioma, melanoma) and adults (liver, brain, cervical) [5]. There has been a 56% increased risk of acute lymphocytic leukemia among children living within two miles of the Ship Channel, compared with children living at least 10 miles away, and residents of neighborhoods nearer to the Ship Channel have more than twice the rate of respiratory disease than other Texans [6,7]. 

As with many other underserved communities in the US, Manchester residents—who are 98% minority, 65% low-income, and 37% living at or below poverty line [8]—lack access to information about the risks of living in proximity to polluting facilities. They frequently seek external partnerships to assist them in fully documenting the wide-ranging implications of these risks [9]. A major petroleum refinery located near the community can produce up to 160,000 barrels of gasoline and other fuels per day. The Interstate Highway 610 Ship Channel Bridge, one of the busiest stretches of Interstate in Houston, carries tens of thousands of vehicles per day (along with their emissions) directly over residences in Manchester [10]. 

In response to these and other documented excess exposure and health inequities, this study aims to use community engaged research and citizen science methods to derive data-driven community master plans to reduce toxic exposure and enhance resilience. Manchester neighborhood residents and affiliated organizations, such as Texas Environmental Justice Advocacy Services (t.e.j.a.s.), have long been involved in community engagement activities around environmental justice issues. Working with local organizations and residents, faculty, engagement staff, and students from the fields of public health, landscape architecture, and urban planning (1) conducted air, dust, and water sampling and analysis within the neighborhood to assess current exposure conditions, (2) developed a master plan to assist in lessening the risks of potential environmental exposures to the community’s residents in the future, and (3) used performance models to project the impacts of the master plan on future flooding and contamination. Through this approach, we seek to determine how citizen science informed environmental and spatial data can be best applied as evidence for design and planning decisions that reduce exposure to flooding and associated contamination during flood events. 

## 2. Materials and Methods 

### 2.1. Engaged Neighborhood Sampling

Located within one mile of 21 facilities that report to the Environmental Protection Agency’s (EPA) Toxic Release Inventory (i.e., 11 large quantity generators of hazardous waste, four facilities that treat, store, or dispose of hazardous wastes, nine major dischargers of air pollution, and eight major storm water discharging facilities), the neighborhood of Manchester, TX is one of many communities adjacent to concentrated areas of industrial land use (CAILU) near the Houston Ship Channel. Manchester is located at the nexus of exposures to hazardous toxic substances and highly socially and physically vulnerable to the impacts of natural disasters, including flooding. Flooding associated with tropical cyclones and inland precipitation events is a major concern for many communities located along the Texas Gulf Coast, including Manchester. According to records, the return period of a normal hurricane along any 50-mile segment of the Texas coast is once every 5 years, with a major hurricane occurring nearly every 15 years [11,12]. However, since 2015, the City of Houston has been impacted by 4 major floods, two of which—Memorial Day Floods (2015) and Tax Day Floods (15 April 2016)—were not associated with a tropical storm and two that were: Hurricane Harvey (2017) and Tropical Storm Imelda (2019). The impacts of all types of floods are being intensified in Houston by urbanization, subsidence, and extreme rainfall [13,14,15]. Relatedly, sea level rise will increase the vulnerability of Texas Gulf Coast communities to hazards such as flooding associated with sea-level rise and increasing storm surge [16]. 

To better understand potential causal pathways between flood exposures and health outcomes and to inform design decisions that may reduce risks, an interdisciplinary group of researchers and designers engaged in a deep participatory process with Manchester residents, stakeholders, and local organizations. Results from a December 2015 community health assessment conducted by the Texas A&M University Institute for Sustainable Communities and t.e.j.a.s. provided support for the hypothesis that hazardous environmental exposures and poor health status were at the forefront of the community’s concerns [17,18]. Therefore, as a citizen science project and in partnership with community stakeholders, especially local high school students, we sampled indoor and outdoor air, indoor dust, surface and drinking water, and surface soil to evaluate the presence and concentrations of toxic exposures in the neighborhood of Manchester and nearby Houston Ship Channel neighborhoods (see Figure 1). 

In November 2016, surface water in 30 public areas across the Manchester neighborhood was sampled and tested for metal(iod)s. These areas were identified by residents as being prone to water pooling after rainfall [18]. Tap water samples were collected from the kitchen faucets of 13 Manchester homes in February 2019 and analyzed for the presence of lead [1]. In December 2016, settled dust samples were collected from a measured and marked one square meter area of hard flooring adjacent to the front door of 25 households using surface wipes [19]. In September 2017, surface soils were collected from the yards of 24 of the homes that were evaluated forsettled dust [19]. Diagnostic polycyclic aromatic hydrocarbon (PAH) ratios were performed to determine the probable sources of PAHs in the indoor settled dust and outdoor yard soil samples. 

In addition to sampling air, water, and soil in Manchester, sampling was conducted in the Ship Channel area following a petrochemical fire. In March of 2019, outdoor air samples (N = 30) were collected from affected public areas to rapidly quantify total volatile organic compounds (TVOC).

Findings from environmental samples highlight a need for additional research to understand the scope of these potential exposures. Research translation was aided by geocoding [20] the location of environmental samples using Geographic Information Systems (GIS). Once locations were mapped and data visualized, we coupled environmental samples with other spatial data. For example, through contour and elevation analyses, we were able to identify divots and depressions in the ground plane with a digital elevation model to identify areas where standing water would be present during inland precipitation events. Sometimes referred to as data inconsistencies, or sinks, these topographic depressions are low points that occur on the land between streams or channels of water and can be mapped through a built-in algorithm within GIS [21]. When performing hydrologic models, if a flow enters a cell lower in elevation than its neighbors (i.e., a sink), sometimes a GIS flow direction algorithm cannot discern where water will flow afterward [22]; then, sinks can be filled to run larger scaled models. However, previous research has demonstrated that filling these areas can be consequential to spatial analyses and overlook important land uses such as wetlands in lieu of predicting larger scaled hydrological flows [23]. In this case, the outputs show homes containing larger concentrations of benzene were in close proximity to existing land depressions or settling points for runoff. These become settling pools of contaminated water that may increase toxic exposure to residents (see Figure 2). 

### 2.2. Participatory Plan Development

The number of participatory urban design projects has increased in underserved communities in recent years, in part to address problems associated with environmental justice [24]. Urban design, in essence, is a problem-solving venture, which makes it well suited to a service-learning or problem-based learning approach [25] In academic settings, service-learning is an alternative teaching model that goes beyond traditional lecture-based teaching to include learning through engagement with community groups to provide students with engaged projects [26,27,28]. Due to the high costs involved with typical urban design projects, marginalized communities can work with university faculty and students to develop conceptual designs that address neighborhood problems and reduce the costs associated with the development of the initial designs [29,30]. Through these types of participatory approaches, faculty and students interact with local residents, community organizations, and other stakeholders [31].

Service-learning projects typically involve a series of collaborative actions with communities, including consultation, information sessions, data gathering, and feedback loops [25,32,33]. This project involved initial presentations about the findings from environmental sampling to inform communities about their potential exposures. Then, the findings were used to develop a master plan that considers land use and environmental changes to assist in the reduction of potential toxic exposure conditions in Manchester, TX. Founded in 1994, t.e.j.a.s works with local residents to provide them with the tools necessary to create sustainable, environmentally healthy, disaster resilient communities. Working in conjunction with t.e.j.a.s, Texas A & M University’s Superfund Research Program, Center for Housing and Urban Development, Texas Target Communities, and Institute for Sustainable Communities involved residents and stakeholders in Manchester to create a community-scaled master plan based on citizen science and deep community engagement. The engagement process used to develop this plan relied on feedback loops that supported design and planning for decreasing potential exposure to toxicants and for improving public health conditions through reduced exposure. Green infrastructure, open space planning and community design scenarios were developed to gather community input through four engagement sessions held over an eight-month period. The master plan design incorporated information from multiple sources to: (1) conduct a site inventory, (2) determine conditions at toxicant-prone areas, (3) develop desired functions for new land uses, and (4) suggest potential infrastructure based on exposure conditions. 

First, an introductory meeting allowed the design team to discuss site-specific problems, initiating a general discussion to help identify high risk areas based on local knowledge, as well as pinpoint current and future flood vulnerable areas. A second meeting presented City of Houston officials with findings from the initial site analysis. Feedback from Manchester residents not only provided further insight that identified unseen conditions but also generated ideas for future land use functions to be incorporated into conceptual design scenarios. A third and fourth meeting involved a feedback loop between community members and design teams in which a series of design scenarios were presented and critiqued by neighborhood residents. Responses from the community to the design team were then used to combine the scenarios into one unified community-scaled master plan. 

### 2.3. Performance Modeling

Quantification of the performance of planning scenarios has become an essential component of flood mitigation [34]. In this study, we used two performance metrics to evaluate the stormwater performance, nonpoint source pollution, and cost-benefit of the plan. The Center for Neighborhood Technology’s (CNT) National Stormwater Management Green Values Calculator, also known as GVC, is a tool that was developed to compare costs, benefits, and performance of development using green infrastructure versus conventional stormwater development practices [35,36]. The GVC has been used to evaluate the performance of green infrastructure in terms of water quality [37], adaptive capacity and flood proofing [38], stormwater runoff capture [39], and stormwater runoff storage [40]. It allows users to specify runoff reduction goals for sites and to input site-specific design parameters including land cover, soil type, runoff reduction goals, and green infrastructure types [41]. Maintenance and construction costs for each green infrastructure type are estimated to provide a life cycle cost for a project at 5-, 10-, 20-, 30-, 50-, and 100-year periods [42].

The Long-Term Hydrologic Impact Assessment (L-THIA) model estimates the average annual runoff and pollutant loads for land use configurations based on more than 30 years of daily precipitation data, soils, and land use data for an area [43]. The L-THIA is an urban growth analysis tool that is applied to estimate long-term runoff and nonpoint source pollution impacts of different land use development scenarios. It provides the estimated long-term average annual runoff, rather than only data for extreme events, based on the use of long-term climate data for specified state and county locations in the U.S. It also generates estimates of 14 types of non-point sources pollutant loadings to waterbodies (e.g., nitrogen, phosphorous, suspended particulates) based on the proposed land use changes. Land use changes in the assessment model include commercial, industrial, high-density residential, low-density residential, water/wetlands, grass/pasture, agriculture, and forest [44]. The model has been used to track land use change in watersheds for historical land-use scenarios [45], identify non-point source pollution sensitive areas, and evaluate land use development for non-point sources pollution management [46].

## 3. Results

### 3.1. Overview of Sampling Outcomes 

Results from environmental sampling and testing found evidence that some pollutants exceeded federal guidelines (Table 1). The concentration of total PAHs in each sample ranged from 0.29 to 3.95 μg/m^2^. Analysis of indoor dust samples identified markers of petroleum in all homes (e.g., C29 Hopane, C30 Hopane, and 18a Oleanane) as well as presence of all 16 priority PAHs and the subgroup of 7 probable human carcinogens. The concentration of total PAHs in each soil sample ranged from 54.7 to 4378.3 μg/kg. Results suggested that the dominant source of both indoor and outdoor PAHs was combustion, which could be attributed to the major transportation infrastructure, 24-line railyard, and industrial complexes that are in close proximity to the neighborhood. Concentrations in excess of the US EPA’s National Recommended Water Quality Criteria for chronic exposure in freshwater aquatic life for arsenic (150 μg/L) and mercury (0.77 μg/L) were found in pooled water samples collected from two locations, while twelve locations exceeded the recommended limit for lead of 3.2 μg/L [47]. Detectable concentrations of barium were also present in water collected from each location sampled, though no recommended limit has been established by the US EPA. Detectable amounts of lead were present in 30.8% (N = 4) at a mean concentration of 1.3 µg/L range: 0.6–2.4 µg/L). Although the US EPA has established a maximum contaminant level goal of zero for lead in drinking water [48], all samples tested were found to be below the regulatory lead action level of 15 µg/L.

### 3.2. Design Outcomes

The design sought to recognize resident concerns related to limiting the dislocation of current residents while simultaneously shield residents from issues related to contaminant collection and transport. Three primary sources of toxicant transferal were identified from the engagement: (1) byproducts from industrial stormwater runoff, (2) hydrologic discharge from the highway overpass, and (3) emissions from railyard traffic and rail deliveries into the neighborhood carrying toxic chemicals for industries (see Figure 3). To address these concerns, first, a riparian zone was injected between the residential area and the industrial zone. This riparian zone comprised a series of strategically placed green infrastructure facilities to contain and clean the byproducts from industrial stormwater runoff during rainstorms. Second, a detention basin was located within the interstitial area in the discharge zone beneath the highway overpass to collect and allow for overflow drainage into appropriate areas away from residential areas. Third, a formal linear park was added along the vacant lots parallel to the railway to separate residential parcels from the potential toxic emissions associated with the railyard’s infrastructure and cargo because, according to residents, trains park for long periods of time to deliver and pick up materials from surrounding industrial sites. Fourth, new housing area that is mixed with proposed small-scaled local commercial land use along the primary arterial was also proposed along the primary arterial. These commercial areas act as a buffer to further separate residential areas from industrial land uses. Finally, new, smaller-scaled low-impact green infrastructure facilities (as opposed to larger scaled ones such as wetlands and urban parks), such as rain gardens, bio-infiltration areas, and vegetated swales, were also proposed to assist with smaller scaled issues related to existing homeowners (see Figure 4). 

### 3.3. Performance of the Plan

To assess the impacts of the master plan and design strategies in pollutant reduction during a flood disaster, we used the Center for Neighborhood Technology’s (CNT) GVC and the L-THIA model that were developed for evaluating the effects of green infrastructure on stormwater runoff and pollutant loads. The GVC was used to assess the performance, costs, and benefits of the green infrastructure utilized within the community-scaled master plan. Table 2 describes the algorithms used by the GVC to determine green benefits. Compared with conventional approaches, the green stormwater best management practices (BMPs) of the design decrease the site’s impervious area by 30%. Simultaneously, the design can capture 147,456 cubic feet of additional runoff, create $331,400 annual green benefit by reducing air pollutants and energy use, provide pollution treatment, increase carbon dioxide sequestration, escalate the compensatory value of trees, and improve groundwater replenishment.

The L-THIA was used to assess the projected performance of the master plan on pollutant totals (See Table 3). Overall, the community-based master plan significantly decreases fecal strep and coliform, as well as suspended solids. Nitrogen, phosphorous, biochemical oxygen demand (BOD), and chemical oxygen demand (COD) are also projected to decrease. Overall, the L-THIA predicts a 41% decrease in 14 pollutants across the site. Specifically, these projections include a 62% reduction of zinc, 59% reduction of chromium, 48% reduction of CODs, 40% reduction of lead, 36% reduction of cadmium, 38% reduction of copper, 32% reduction of nickel, 29% reduction of BODs, 22% reduction of phosphorous, and a 4% reduction of nitrogen. Simultaneously, the total annual runoff amount and volume were significantly decreased, while new housing options increase diversity in residential homesteads.

## 4. Discussion

This project sought to improve our understanding of how citizen science-informed spatial data could be combined with limited data available from environmental sampling and landscape performance calculators to provide evidence for design and planning decisions that reduce exposure to flooding and flood-mobilized contaminants among residents of a Houston environmental justice community. Developing new approaches to mitigate the impacts of flooding will become more important because, as the frequency and severity of both nuisance and major flooding increases globally, the threats to human and environmental health, particularly in CAILU neighborhoods, will also intensify. For example, even relatively small increases in the global mean sea level are predicted to double the frequency of both coastal and tidal flooding over the next several decades [49,50]. Increased human activities such as mining, dam building, and the pumping of water or natural gas are increasing subsidence and intensifying inland flooding [14,51]. The risks from natural hazards such as flooding will continue to be disproportionately felt among socially vulnerable groups who reside in environmental justice communities [52].

A major methodological contribution of this research is the integration of primary environmental assessments with pre-design analysis, design development, and post-design evaluation. Existing (secondary) data are often too coarse in spatiotemporal resolution to address context-specific problems and inform actionable design decisions. The field sampling identified site-specific issues, pinpointed the spatial extent, quantified the severity of different pollutants in the neighborhood, and suggested possible environmental risk factors and infrastructure deficiencies that could be addressed in the design phase. Guided by this fine-grained data, the master plan is developed to target specific hazardous conditions and toxic materials. The performance evaluation process allows different design alternatives to be compared and the post design reductions of the pollutants to be quantified. Such a process ensures quantifiable metrics are truly integrated into the design and enhances the rigor of community engaged design research that has inherent subjectivity.

In addition to the quantifiable assessments, this project also demonstrates how service-learning programs can be used to integrate community-sourced data into interdisciplinary design projects and used for design decision making. In this case, the deep community engagement enabled by linkages between communities and a group of centers, institutes, and programs at Texas A & M University allowed for the alignment of a design program with the community’s self-identified needs. This meant that projects could have a stronger positive impact on annual economic benefits, stormwater retention, and contaminant reduction. Citizen scientists served several important roles that supported these outputs. First, citizen science informed the rationale for design decision making. Too often, designers and planners simply rely on aesthetics and existing frameworks to develop master plans. The use of engagement and locally generated evidence to reinforce decision making clearly produces stronger, and more relevant results. Second, it focused the plan by identifying specific issues of priority to the community that could be directly treated through design. This provided a series of targeted actions for which strategies could be developed to solve. Third, the dissemination of findings from our research helped improve the community’s understanding of the causes and effects of environmental contamination. This form of educating residents of CAILU neighborhoods through visualization and presentation of results works as part of a feedback loop and allows communities to make better decisions through engagement. Fourth, participation in this process—and the related implicit valuation of local knowledge—spilled over into other aspects of engagement and assisted the research team in understanding conditions that we were unable to see by not living, working, and socializing in the neighborhood on a daily basis. Finally, the products generated through these processes were able to both identify and suggest provisions for community toxic transferal issues and provide hope moving forward. The documents created can be used by neighborhood advocates to support their case to policy makers to help improve conditions in the future. 

This approach is not without limitations. First, both the research team and the community must be both flexible and highly dedicated to the project since community engaged design can take longer to complete than a design that is not informed by local knowledge, deep local involvement, and multiple opportunities for feedback. Community timelines are not based on the academic semester but are generated by their own priorities, meaning that community members may not always available to participate in research activities on an academic timeline. When working with a diverse set of community stakeholders, research and engagement staff must be prepared for the reality that the concerns and priorities of community stakeholders will not be uniform and may in fact be at odds with one another. University faculty, staff, and students often require cultural competency training to break down assumptions and address stereotypes. Therefore, working with experienced engagement staff like those at the University’s Center for Housing and Urban Development, Texas Target Communities, and Institute for Sustainable Communities who have developed trust with partner communities is critical to ensure that neighborhood representatives feel comfortable and empowered in sharing information. In the long-term, ongoing engagement supported through institutional resources that allow for sustained communication between community leaders and academic partners ensures more credible inputs and fosters more reliable outputs. Engagement partners are more hesitant to engage with researchers in short-term or one-off projects that operate based on a grant funding cycle since they are unlikely to realize real change on those externally driven timelines. If these challenges can be addressed and participatory activities adequately prepared for, citizen involvement can improve chances for success of this type of sustainability research while accomplishing dual goals that extend scientific knowledge and build local capacity to enacting positive change within communities. 

## 5. Conclusions

Our study offers new insights into addressing contaminant exposure of vulnerable communities and tests a methodological framework that integrates community engaged research, citizen science environmental assessments, and community plan development using a CAILU neighborhood prone to frequent flood (such as Manchester, TX) as a testbed. The research is novel in that it combines the interdisciplinary fields of public health, landscape architecture, and urban planning with local-scaled citizen science to solve issues related to contamination and flooding. The research aims to develop actionable plans through land use and environmental modifications to transform CAILU neighborhoods through reduction of toxic exposure and flooding. The research approach can easily be applied to other neighborhoods experiencing similar circumstances, as these are ubiquitous problems across many urban areas. To achieve this, it is important to develop trust and long-term relationships with local community members and organizations across transdisciplinary fields of research and allow the communities to understand what each field can help them achieve. Specific tasks need to be developed for each discipline, with a focus on how each can build off and integrate with one another. 

Our findings demonstrate the importance of environmental sampling data and community engagement processes to inform planning and design. As a result of these processes, the plan achieves the goals of reducing runoff, air and water pollutions, increasing carbon dioxide sequestration, and facilitating groundwater replenishment. The plan also leads to an average of 41% decrease across all analyzed pollutants, reducing potential exposure to and transferal of toxic materials. Taken together, our findings illustrate the strength of community engaged research to identify environmental features important to residents and create action-oriented solutions towards environmental justice, health, and resilience. 

## Figures and Tables

**Figure 1 ijerph-17-00486-f001:**
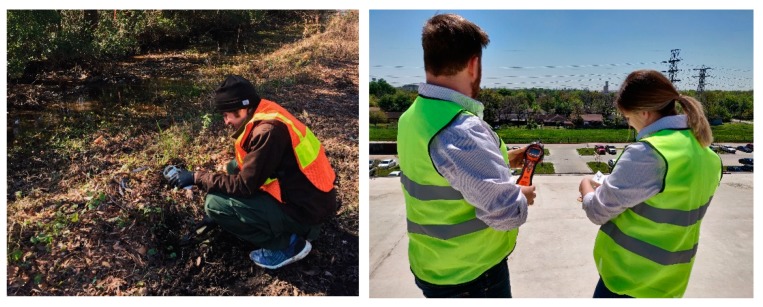
Water (**left**) and air (**right**) quality sampling in Manchester, TX.

**Figure 2 ijerph-17-00486-f002:**
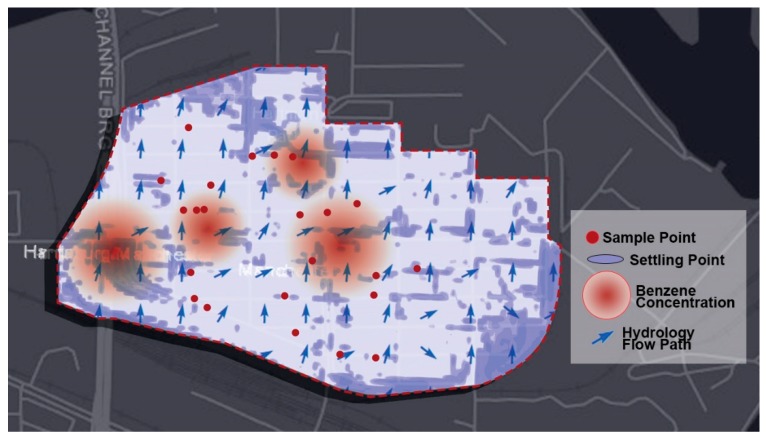
Geocoded sampling points, water settling points, and major benzene concentrations in Manchester, TX.

**Figure 3 ijerph-17-00486-f003:**
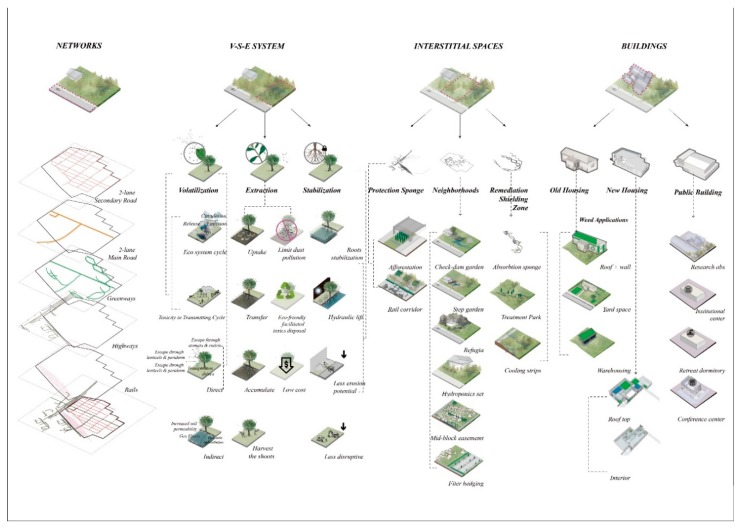
Schematics and program of the participatory master plan developed for Manchester, TX.

**Figure 4 ijerph-17-00486-f004:**
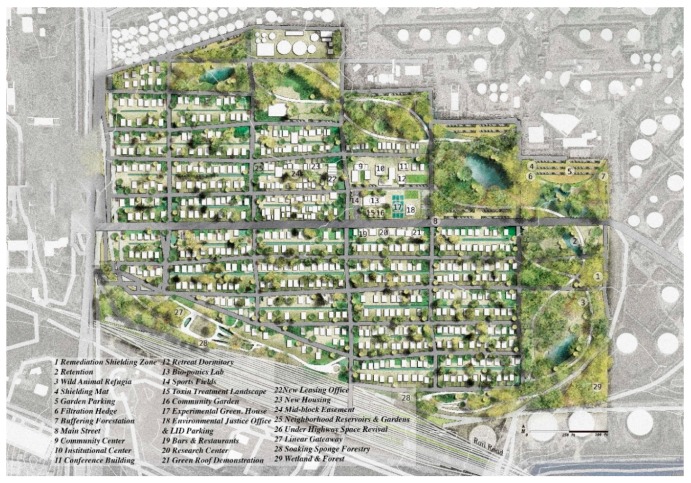
Master plan developed through deep engagement for Manchester neighborhood.

**Table 1 ijerph-17-00486-t001:** Overview of environmental sampling in Houston Ship Channel neighborhoods, 2015–2019.

Location	Sample Date	Sample Type	Summary of Findings
Manchester	Dec. 2016	Indoor Settled Dust (N = 25)	The total concentration of PAHs in each sample of household dust ranged from 0.9 to 11.1 mg/m^2^ with a respective mean and median of 2.3 and 1.6 mg/m^2^. In addition, markers of petroleum (e.g., C29 Hopane, C30 Hopane, and 18a Oleanane) were present in all household dust samples.
Manchester	March 2017	Indoor Air (N = 11)	Multiple compounds present in samples that are regulated as hazardous air pollutants by US EPA. These included Acetone, Benzene, Trichloroethene, Pentanal, Toluene, Tetrachloroethene, Acetic Acid Butyl Ester, Hexanal, Ethylbenzene, m-Xylene, o-Xylene, Styrene, Benzene, Propyl-, Benzene 1 ethyl 3 methyl, Benzene 1 ethyl 2 methyl, 1,3,5-Trimethylbenzene, Benzene, 1,2,4-trimethyl-, Benzaldehyde, Benzene, 1-methyl-4-(1-methyl ethyl), Benzene, 1,4-dichloro, Phenol, Hexachlorobutadiene, and Naphthalene. Pentanal, Acetic Acid Butyl Ester, Hexanal, Benzene 1 ethyl 3 methyl, Benzene 1 ethyl 2 methyl, Benzene, 1-methyl-4-(1-methyl ethyl), and Phenol.
Manchester	Sept. 2017	Outdoor Yard Soils (N = 24)	The concentration of total PAHs in each soil sample ranged from 54.7 to 4378.3 μg/kg
Manchester	November 2016	Pooled Surface Water	Arsenic was detected in eight samples at a mean concentration of 54 μg/L (range: 10–180 μg/L), chromium was detected in 10 samples at a mean concentration of 65 μg/L (range: 11–363 μg/L), and lead was detected in 12 samples at a mean concentration of 195 μg/L (range: 17–1448 μg/L). Mercury was detected at a level of 10 μg/L in two samples, while barium was detected in all 30 samples at an average level of 306 μg/L (range: 46–3296 μg/L).
Manchester	Feb. 2019	Tap Water (N = 13)	Of these, 30.8% were found to be positive for lead at levels ranging from 0.6 to 2.4 µg/L. Although the US EPA has established a maximum contaminant level goal of zero for lead in drinking water [48], all samples tested were found to be below the regulatory lead action level of 15 µg/L.
Houston Ship Channel Area	March 2019	Outdoor Air (N = 30)	Concentrations of TVOC ranged from below the limit of detection of 0.001 ppm to 3.3 ppm, and the maximum level of benzene detected was 0.1 ppm.

**Table 2 ijerph-17-00486-t002:** Green infrastructure benefit quantification and valuation equations for the GVC.

Benefit Type	Reference	Equations
Reducing stormwater runoff	Equation (1):	Total runoff reduction (gal) = [annual precipitation (inches) × GI area (SF) × % retained] × 144 sq inches/SF × 0.00433 gal/cubic inch
	Equation (2):	Avoided stormwater treatment costs ($) = runoff reduced (gal) × avoided cost per gallon ($/gal)
Improving air quality	Equation (3):	Total annual air pollutant uptake/deposition (lbs) = area of practice (SF) × average annual pollutant uptake/deposition (lbs/SF) Where annual pollutant uptake/deposition (lbs/SF) ranges from 3.00 × 10^−4^ lbs/SF/yr to 4.77 × 10^−4^ lbs/SF/yr (Currie and Bass 2008; Yang, Qian, and Gong 2008) Total annual criteria pollutant reduction benefit (lbs) = ∑total criteria pollutant uptake/deposition benefit (lbs) + Total avoided criteria pollutant emissions (lbs)
Reducing atmospheric CO2	Equation (4):	Total annual climate benefit (lbs CO_2_) = ∑total equivalent sequestration benefit (lbs CO_2_) + total avoided CO_2_ emissions (lbs CO_2_)
	Equation (5):	Total annual value of climate benefit ($) = total climate benefit (lbs CO_2_) × price of CO_2_ ($/lb)
Return/ Time	Equation (6):	Total GI annual benefit ($) × Years = Total GI construction cost ($) + Total GI annual maintenance cost ($) × Years
	Equation (7):	Total GI annual benefit ($) × Years = Total neighborhood construction cost ($) + Total neighborhood annual maintenance cost ($) × Years

**Table 3 ijerph-17-00486-t003:** Long-Term Hydrologic Impact Assessment (L-THIA) Low Impact Development (LID) spreadsheet inputs and outputs measuring master plan performance.

**Summary**				
**Land Use**	**Hydrologic Soil Group**	**Pre-Developed**	**Post-Developed W/o LID**	**Post-Developed with LID as Proposed**
HD Residential 1/8 acre	C	94.18	28.54	15.69
Industrial	C	4.46	4.09	-
LD Residential 1/2 acre	C	53.21	32.24	20.95
Grass/Pasture	C	55.17	130.46	-
Commercial	C	-	5.99	2.39
Water	C	-	5.7	-
**Percentage Impervious**				
**Land Use**	**Default**		**Adjusted**	
Residential 1/4 acre	38		-	
Residential 1/8 acre	65		65	
Residential 2 acre	12		-	
Residential 1 acre	20		-	
Residential 1/2 acre	25		25	
Commercial	85		85	
Industrial	72		72	
**Avg. Annual Runoff Volume (Acre-Ft)**				
**Land Use**	**Current**		**Post-Developed with LID as Proposed**	
HD Residential 1/8 acre	103.57		29.94	
Industrial	5.21		4.77	
LD Residential 1/2 acre	29.67		17.98	
Grass/Pasture	20.42		48.29	
Commercial	0		9.01	
Water	0		0	
Total Annual Volume (acre-ft)	158.88		110.01	
**Nonpoint source pollutant results**				
	**Pre-Developed**		**Post-Developed with LID as Proposed**	
Nitrogen (lbs)	715		377	
Phosphorous (lbs)	209.556		84	
Suspended Solids (lbs)	15,799		7633	
Lead (lbs)	3.218		2.345	
Copper (lbs)	3.496		2.725	
Zinc (lbs)	31.333		16.789	
Cadmium (lbs)	0.354		0.277	
Chromium (lbs)	1.277		1.595	
Nickel (lbs)	2.925		1.701	
Biochemical Oxygen Demand (BOD) (lbs)	9483		4141	
Chemical Oxygen Demand (COD) (lbs)	18,617		9905	
Oil & Grease (lbs)	658		481	
Fecal Coliform (millions of coliform)	33,681		13,333	
Fecal Strep (millions of coliform)	92,809		35,607	

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
