# Peer review of "Citizen Science-Informed Community Master Planning: Land Use and Built Environment Changes to Increase Flood Resilience and Decrease Contaminant Exposure"

_ijerph, 2020, doi:10.3390/ijerph17020486_

Round 1

Reviewer 1 Report

The research developed actionable plans coupling land use and environmental modifications to transform CAILU neighborhoods by reducing risks of toxic exposure and flooding. This study proposes a “research for design” method, which is interesting and meaningful. Also it is important for landscape designers. I suggest accepting this manuscript after minor revisions:

--I suggest the authors reconsider the title. The current title does not fully reflect the core theme of the study.

--Abstract: line 23-’147,456 ft3 of...’  what does 3 mean?

--The novelty of this study should be highlighted.

--I suggest the authors add a description on how to apply this method on other similar cases, or its  practical significance.

Author Response

The research developed actionable plans coupling land use and environmental modifications to transform CAILU neighborhoods by reducing risks of toxic exposure and flooding. This study proposes a “research for design” method, which is interesting and meaningful. Also it is important for landscape designers. I suggest accepting this manuscript after minor revisions:

--I suggest the authors reconsider the title. The current title does not fully reflect the core theme of the study.

The title has been adjusted to “Citizen Science-Informed Community Master Planning: Land Use and Built Environment Changes to Increase Flood Resilience and Decrease Contaminant Exposure.”

--Abstract: line 23-’147,456 ft3 of...’ what does 3 mean?

This was referencing 3 cubic feet. The text has been updated to better reflect this.

--The novelty of this study should be highlighted.

The novelty has been highlighted in a paragraph in the conclusions

--I suggest the authors add a description on how to apply this method on other similar cases, or its  practical significance.

This has been added in the conclusions as well and linked to the novelty.

Reviewer 2 Report

This manuscript summarizes a case study conducted in Manchester, TX that engaged the affected community in development of a project to alter land use and build changes in the environment that would increase resilience to flood and decrease exposure to contaminants. This is important work that provides evidence that such an approach is effective and should be showcased as an example for others.

Critique

Although I find the work to commendable and effective, the manuscript is ponderous in several places. The problem is that the authors write overly long sentences that often use strings of gerunds whose placement makes sentences difficult to discern. This issue is not always their fault; some of the terms of art (e.g., built environment changes) when placed in sentences are the cause of ambiguity. The authors are requested to look at their manuscript carefully in order to divide longer compound sentences into shorter simple sentences and to attempt to reduce the use of sequential gerunds where possible.

Specific Comments

Title: The title is difficult to understand. A possible new title is: :Citizen Science-Informed Community Master Plan: Land Use and Built Environment Changes to Increase Flood Resilience and Decrease Contaminant Exposure.”

Line 66: please change “on” to “one”

Line 148: “uses” is repeated two words later. Perhaps change “uses” to “considers”

Line 166: Consider changing “an initial site analysis” to “the initial site analysis”

Lines 166-167: Consider rewording as “…Manchester residents not only provided further insight that identified unseen conditions but also generated ideas…”

Line 178: Consider changing “compared to” to “versus”

Line 212: Please change to read “…locations exceeded the recommended limit…”

Line 214: Consider changing “equivalent limit” to “recommended limit”

Line 229: Consider altering to read “…industrial zone. This riparian zone comprised a series of strategically…”

Line 235: Consider altering to “infrastructure and cargo because, according to residents,…”

Lines 237-238: Consider rewording as “…new housing area that is mixed with proposed small-scaled local commercial land use along the primary arterial.”

Line 241: The meaning of “ smaller scaled issues related to existing homeowners” is unclear. Were these issues identified by homeowner? If so, please replace “related to” with “identified by.” If not, please make appropriate changes to improve clarity.

Line 271: The word “Specially” seems incorrect. Do you mean “Specifically”?

Line 275: Please change “was” to “were” (plural subject) Please note the tenses on the clauses do not agree. “Were” is past tense, but “increase” is present tense.

Line 285: Consider changing “…important since as the frequency…” to …Important because, as the frequency…”

Line 319” Consider replacing “presentation of data findings” to “presentation of results”

Line 330: Consider changing “deep engagement” to “deep involvement” (engage is used in the preceding line of the same sentence and sounds redundant)

Line 335: Consider changing “monolithic” to “uniform” or ‘Homogeneous”

Lines 353-354: Consider rewording as: “…community plan development using a CAILU neighborhood prone to frequent flood (Manchester, TX) as a testbed”

Line 358: Consider rewording as “…engagement process to inform planning and design”

Line 363: The word “pathogenic” has medical implications and seems inappropriate for community involvement. Perhaps change wording to “…community-engaged research in identifying environmental features important to the residents…”

Author Response

community in development of a project to alter land use and build changes in the environment that would increase resilience to flood and decrease exposure to contaminants. This is important work that provides evidence that such an approach is effective and should be showcased as an example for others.

Critique

Although I find the work to commendable and effective, the manuscript is ponderous in several places. The problem is that the authors write overly long sentences that often use strings of gerunds whose placement makes sentences difficult to discern. This issue is not always their fault; some of the terms of art (e.g., built environment changes) when placed in sentences are the cause of ambiguity. The authors are requested to look at their manuscript carefully in order to divide longer compound sentences into shorter simple sentences and to attempt to reduce the use of sequential gerunds where possible.

A substantial edit has been made for clarity

Specific Comments

Title: The title is difficult to understand. A possible new title is: :Citizen Science-Informed Community Master Plan: Land Use and Built Environment Changes to Increase Flood Resilience and Decrease Contaminant Exposure.”

The title has been adjusted to “Citizen Science-Informed Community Master Planning: Land Use and Built Environment Changes to Increase Flood Resilience and Decrease Contaminant Exposure.”

Line 66: please change “on” to “one”

The text has been updated based on the reviewer’s comment.

Line 148: “uses” is repeated two words later. Perhaps change “uses” to “considers”

The text has been updated based on the reviewer’s comment.

Line 166: Consider changing “an initial site analysis” to “the initial site analysis”

The text has been updated based on the reviewer’s comment.

Lines 166-167: Consider rewording as “…Manchester residents not only provided further insight that identified unseen conditions but also generated ideas…”

The text has been updated based on the reviewer’s comment.

Line 178: Consider changing “compared to” to “versus”

The text has been updated based on the reviewer’s comment.

Line 212: Please change to read “…locations exceeded the recommended limit…”

The text has been updated based on the reviewer’s comment.

Line 214: Consider changing “equivalent limit” to “recommended limit”

The text has been updated based on the reviewer’s comment.

Line 229: Consider altering to read “…industrial zone. This riparian zone comprised a series of strategically…”

The text has been updated based on the reviewer’s comment.

Line 235: Consider altering to “infrastructure and cargo because, according to residents,…”

The text has been updated based on the reviewer’s comment.

Lines 237-238: Consider rewording as “…new housing area that is mixed with proposed small-scaled local commercial land use along the primary arterial.”

The text has been updated based on the reviewer’s comment.

Line 241: The meaning of “ smaller scaled issues related to existing homeowners” is unclear. Were these issues identified by homeowner? If so, please replace “related to” with “identified by.” If not, please make appropriate changes to improve clarity.

The text has been updated based on the reviewer’s comment.

Line 271: The word “Specially” seems incorrect. Do you mean “Specifically”?

The text has been updated based on the reviewer’s comment.

Line 275: Please change “was” to “were” (plural subject) Please note the tenses on the clauses do not agree. “Were” is past tense, but “increase” is present tense.

The text has been updated based on the reviewer’s comment.

Line 285: Consider changing “…important since as the frequency…” to …Important because, as the frequency…

The text has been updated based on the reviewer’s comment.

Line 319” Consider replacing “presentation of data findings” to “presentation of results”

The text has been updated based on the reviewer’s comment.

Line 330: Consider changing “deep engagement” to “deep involvement” (engage is used in the preceding line of the same sentence and sounds redundant

The text has been updated based on the reviewer’s comment.

Line 335: Consider changing “monolithic” to “uniform” or ‘Homogeneous”

The text has been updated based on the reviewer’s comment.

Lines 353-354: Consider rewording as: “…community plan development using a CAILU neighborhood prone to frequent flood (Manchester, TX) as a testbed

The text has been updated based on the reviewer’s comment.

Line 358: Consider rewording as “…engagement process to inform planning and design”

The text has been updated based on the reviewer’s comment.

Line 363: The word “pathogenic” has medical implications and seems inappropriate for community involvement. Perhaps change wording to “…community-engaged research in identifying environmental features important to the residents…”

The text has been updated based on the reviewer’s comment.

Reviewer 3 Report

This project sought to improve our understanding of how citizen science-informed spatial data 280 could be combined with limited data available from environmental sampling and landscape 281 performance calculators to provide evidence for design and planning decisions.

The paper is well organized, with good idea and excellent design.
the key point is how to measure neighborhood proximity and Resiliency using scientific method.

Author Response

This project sought to improve our understanding of how citizen science-informed spatial data could be combined with limited data available from environmental sampling and landscape performance calculators to provide evidence for design and planning decisions. The paper is well organized, with good idea and excellent design. The key point is how to measure neighborhood proximity and Resiliency using scientific method.

Thank you for your feedback. We had adjusted the manuscript based on all reviewer’s comments.

Round 2

Reviewer 3 Report

The topic is more eye-catching and the structure is reasonable, so it is suggested to publish.